# Dual Microcapsules Encapsulating Liquid Diamine and Isocyanate for Application in Self-Healing Coatings

**Huaixuan Mu** [1], **Yiqing Deng** [2], **Wangcai Zou** [1], **Xiandi Yang** [1] **and Qiang Zhao** [1,*]

[1] School of Chemical Engineering, Sichuan University, Chengdu 610065, China; mhx@stu.scu.edu.cn (H.M.); zouwangcai1234@163.com (W.Z.); 13987429215@163.com (X.Y.)
[2] School of Polymer Science and Engineering, Sichuan University, Chengdu 610065, China; dengyiqing@stu.scu.edu.cn
[*] Correspondence: zhaoqiang@scu.edu.cn; Tel.: +86-151-8432-5181

**Abstract:** Dual microcapsule systems, especially those based on the polyurea matrix, have emerged as pivotal components driving innovation in self-healing materials, thanks to the intrinsic properties of polyurea, primarily diamine and diisocyanate, rendering it an optimal choice for enhancing self-healing coatings. However, the encapsulation of polyurea components is fraught with substantial technical hurdles. Addressing these challenges, a novel methodology has been devised, leveraging n-heptane as a solvent in the liquid diamine emulsion process to facilitate the synthesis of diamine microcapsules. These microcapsules exhibit a uniform spherical morphology and a robust shell structure, with an encapsulated core material ratio reaching 39.69%. Analogously, the encapsulation process for diisocyanate has been refined, achieving a core material percentage of 10.05 wt. %. The integration of this bifunctional microcapsule system into diverse polymeric matrices, including epoxy resins and polyurethanes, has been demonstrated to significantly enhance the self-healing efficacy of the resultant coatings. Empirical validation through a series of tests, encompassing scratch, abrasion, and saltwater immersion assays, has revealed self-healing efficiencies of 21.8% and 33.3%, respectively. These results indicate significant improvements in the durability and self-repair capability of coatings, marking a notable advancement in self-healing materials with promising potential for tailored applications in automotive, aerospace, and construction industries.

**Keywords:** dual microcapsules system; self-healing; polyurea; coatings; interfacial polymerization





## 1. Introduction

In the realm of contemporary materials science, self-healing coating technology has attracted significant attention [1–5]. This attention is not only due to its considerable benefits in extending material service life and reducing maintenance costs but also to its potential to enhance structural integrity and safety [6–8]. Traditional materials frequently suffer from wear, tear, and cracking under prolonged use or extreme conditions, leading to performance degradation and a reduced lifespan [9–11]. Self-healing coating technology, which mimics the self-repair capabilities of biological systems, can undergo self-repair upon stimulation, thereby significantly enhancing material durability and reliability [12–14].

Self-healing materials are principally classified into two types: intrinsic and extrinsic [15]. Intrinsic self-healing materials facilitate self-repair through reversible chemical reactions within the material, including covalent bonds, hydrogen bonds, and metal–ligand complexes [16]. Conversely, extrinsic self-healing materials, such as liquid-core fibers and microcapsules, have found broader industrial applications [17]. Microcapsules, acting as containers for healing agents, exhibit greater utility due to their capacity to detect micro-cracks and their ease of encapsulation [18].

Since the invention of carbonless copy paper in 1954 [19], microcapsule technology has found extensive application in self-healing materials [20]. This technique entails encapsulating healing agents within tiny capsules [21–23]. When cracks form in the material, the

expanding fissures cause the microcapsules to rupture, enabling the healing agent to flow to the crack site via capillary action and undergo a polymerization reaction catalyzed by a catalyst, thus repairing the crack [24]. The repair reaction must satisfy specific conditions: the healing agent must be a low-viscosity liquid with high wettability to the fracture surface and amenable to encapsulation in microcapsules; the polymerization catalyst (curing agent) must not react with the matrix or be capable of encapsulation within the microcapsules; and the healing agent must polymerize at room temperature in the presence of the catalyst, yielding a product with robust mechanical properties and adhesive strength [25–27]. The advantages of this method encompass its efficiency, controllability, and minimal interference with the material's intrinsic properties. In recent years, microcapsule technology has seen broad application across various fields, including medicine [28,29], coatings [30–33], food [34–37], and agriculture [38,39]. However, the slow repair rate of microcapsules [40], the selection of healing agents [21,25,41], and the development of effective encapsulation [42–45] and release control methods [46–49] continue to be major research focuses and challenges.

Liquid diamine and isophorone diisocyanate (IPDI) are promising candidates for core materials of microcapsules, especially in dual self-healing systems. When kept separate at room temperature, these materials exhibit considerable flowability and chemical stability. However, upon contact, IPDI rapidly reacts with diamine to form and solidify polyurea, a swift, clean, and pollution-free process that does not release non-condensable gases. The resultant polyurea possesses excellent chemical stability, high elasticity, and corrosion resistance, effectively preventing secondary crack propagation. By encapsulating these highly reactive compounds, resin materials, including epoxy resins and polyurea, can be imparted self-healing capabilities. This not only broadens the application spectrum of high-reactivity polyurea but also introduces new, rapid-reacting options for healing agents in microcapsule-based self-healing materials.

The implementation route of this study comprised several key steps. First, the emulsification system for the liquid diamine was determined. Owing to its stability at room temperature and solubility in most water and organic solvents [50], the microencapsulation process presented technical challenges. In this study, n-heptane was chosen as the dispersion phase for emulsification. As a stable organic solvent, n-heptane exhibits good immiscibility with diamine and volatility, simplifying microcapsule post-treatment and reducing potential reactions between the isocyanate and water. To the best of our knowledge, this represents the first use of n-heptane as an emulsion system. Subsequently, the microcapsules were prepared using the interfacial polymerization method to encapsulate diamine and isocyanate within a polyurea shell. This method offers a high recovery rate of microcapsules, indicating its potential for industrial-scale production. Lastly, widely used commercial epoxy resins and polyurea resins (PU-F520), similar to the shell materials of microcapsules, were selected for use in the study, and their self-healing performance were tested to ensure comprehensive and reliable experimental results. By simulating various damage scenarios and assessing their self-healing efficiency, we delved into the practicality of the dual-component microcapsule system for industrial applications.

In this paper, a novel method for microencapsulating liquid diamine and IPDI is introduced. These microcapsules are proposed as additives to confer self-healing properties on coatings. When the coating is subjected to stress, the microcapsules at the site of damage rupture, releasing the polyurea monomer components. These agents are subsequently transported to the crack site through capillary action, where they commence the repair process. Moreover, the small size effect [51] of the microcapsules serves to block the ingress of air and water, significantly augmenting the corrosion resistance of the coatings. Additionally, this study expands the emerging application field of polyurea as a rapid-reacting, two-component resin, offering a new approach for polyurea usage and providing innovative options for healing agents in self-healing materials.

## 2. Experimental

### 2.1. Materials

Ethylenediamine (EDA), sodium dodecylbenzenesulfonate (SDBS), and Tween 80 were procured from Chengdu Cologne Chemical Co., Ltd. (Chengdu, China). 1,4-Cyclohexanediamine (HMDA) and m-phenylenediamine (MXDA) were obtained from Shanghai Macklin Biochemical Co., Ltd. (Shanghai, China), and diethylenetriamine (DETA), polyetheramine D230 (PEA-D230), and isophorone diisocyanate (IPDI) were sourced from Shanghai Aladdin Biochemical Technology Co., Ltd. (Shanghai, China). Three types of resin materials, namely epoxy resin and aspartic polyurea resin F520, were sourced from Shanghai MEGA Adhesive Products Co., Ltd. (Shanghai, China) and Shenzhen Feiyang Protech Corp., Ltd. (Shenzhen, China), respectively.

### 2.2. Synthesis of Diamine Microcapsules

Through the water-in-oil interfacial polymerization technique, diethylenediamine microcapsules were successfully synthesized (Scheme 1a). These microcapsules encapsulate a liquid diethylenediamine core, surrounded by a robust polyurea shell formed through the reaction with IPDI. For example, in preparing microcapsules with an ethylenediamine core, a specified quantity of ethylenediamine (0.015 mol) is mixed with 10 mL of n-heptane to form a diethylenediamine emulsion, using SDBS (1 wt. %) as the emulsifier. The reaction occurs at room temperature. The diethylenediamine emulsion is then incrementally added dropwise at a rate of one drop per second to 30 mL n-heptane solution containing IPDI (0.25 mol/L), with continuous stirring for 2 h to solidify the spherical structure. The ethylenediamine microcapsules are then obtained through washing, filtering, and drying at room temperature. Microcapsules with 1,4-diaminocyclohexane and m-xylylenediamine as core materials were also prepared, named sequentially as MCs-EDA, MCs-HMDA, and MCs-MXDA.

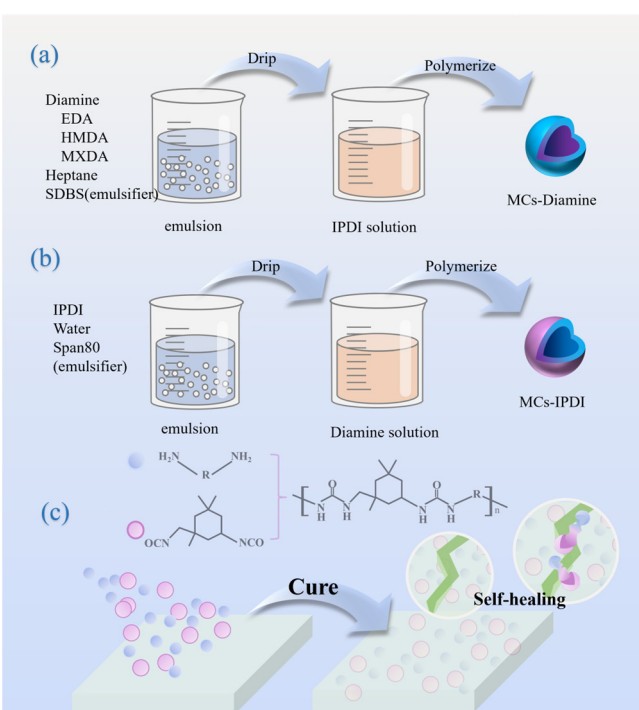

**Scheme 1.** Synthesis mechanism illustration: (**a**) MCs-Diamine; (**b**) MCs-IPDI; (**c**) self-healing coatings.

### 2.3. Synthesis of IPDI Microcapsules

The synthesis of IPDI microcapsules, labeled MCs-IPDI, closely follows the process used for MCs-Diamine, utilizing an oil-in-water emulsion (Scheme 1b). The aqueous phase comprises deionized water, with Tween80 acting as the emulsifier. The specific experimental procedure is as follows: 6 g IPDI is emulsified with 30 mL of water to create an emulsion,

utilizing 1 wt. % Tween80. A quantity of 9.0 mmol diamine is dissolved in 30 mL of deionized water to form a diamine solution. The IPDI emulsion is then gradually added to the solution at a rate of one drop per second to produce microcapsules, with continuous stirring for 2 h to solidify the microcapsule shell and prevent agglomeration. The reaction occurs at room temperature. The microcapsules are then collected by filtration, washed with deionized water, and air dried at room temperature for 48 h to obtain microcapsule powder. Various diamines were utilized to synthesize the isocyanate microcapsules, yielding types named MCs-IPDI(EDA), MCs-IPDI(DETA), and MCs-IPDI(D230).

### 2.4. Preparation of Self-Healing Coatings

In this study, two typical resins were selected for preparing self-healing coatings: epoxy resin and aspartic polyurea (Scheme 1c). These resin materials were formulated in certain proportions, and each was doped with two types of microcapsules at an equal molar ratio (1:1). The specific preparation processes are as follows:

Epoxy Resin Preparation: Components A and B were mixed in a mass ratio of 1:1. The effective reaction of the epoxy resin components was ensured to form a uniform material structure.

Aspartic Polyurea Preparation: Prepared by mixing amine and isocyanate components in a 1:1 molar ratio. The amine component was F520, and the isocyanate component was a homemade IPDI prepolymer. These two components were mixed in a 1:1 molar ratio to ensure the equilibrium of the reaction and the consistency of the product.

The prepared resins were subsequently poured into polytetrafluoroethylene molds for shaping. To eliminate bubbles and ensure resin uniformity, degassing was conducted in a vacuum environment, followed by room temperature curing. This process not only stabilizes the resin's form but also ensures the even distribution of microcapsules within the resin, thereby enhancing the self-healing coating's performance.

### 2.5. Characterization of Microcapsules

The surface morphology of the microcapsules was examined using optical microscopy and scanning electron microscopy (SEM, Apreo S HiVac, Thermo Scientific, Waltham, MA, USA), while their core–shell structure was verified using transmission electron microscopy (TEM, Talos F200i, Thermo Scientific, Waltham, MA, USA). The chemical composition of the microcapsules was identified using infrared spectroscopy. Their performance and core material activity were further evaluated using thermogravimetric analysis (TGA) and differential scanning calorimetry (DSC) in a nitrogen atmosphere, with a heating rate of 10 °C/min from 0 °C to 800 °C, using a simultaneous TG-DSC analyzer (METTLER 1100LF, Mettler Toledo, Urdorf, Switzerland). The particle size analysis of the microcapsules was conducted using Nano Measure software 1.2, with 100 samples chosen for analysis based on the SEM images obtained [43].

The core content of diamine microcapsules was determined in accordance with ATSM D2074-07(2019) standards. The microcapsules were dissolved in a solvent (such as ethylene glycol) to release their core material, then three drops of bromophenol blue indicator were added and titrated to the endpoint with a hydrochloric acid–isopropanol standard solution.

The core content of diamine microcapsules is expressed as a mass percentage, calculated as follows:

$$w(Diamine)\% = \frac{0.5 \cdot c_{HCl} \cdot V \cdot M}{m} \times 100\% \tag{1}$$

Here, $c_{HCl}$ represents the concentration of the standard hydrochloric acid solution (calibrated using anhydrous sodium carbonate), $V$ denotes the volume of the consumed HCl standard solution, $M$ is the relative molecular mass of the diamine used in the sample, and m is the mass of the sample.

The core content test for isocyanate microcapsules was performed using a di-n-butylamine-anhydrous toluene titration method, as per HG/T 20679-2017 standards. The core material from the microcapsules was dissolved in a solvent and then 25 mL of anhy-

drous toluene was added to a conical flask and shaken until completely dissolved. Next, 25 mL of di-n-butylamine-anhydrous toluene solution was added and reacted uniformly for 15 min, followed by the addition of 100 mL of isopropanol. After adding 4–6 drops of bromophenol blue indicator, the mixture was titrated to the endpoint with a hydrochloric acid standard solution.

The content of the isocyanate base (*NCO*%) is expressed as a percentage by mass and is calculated using the following formula:

$$NCO\% = \frac{(V_0 - V) \cdot c_{HCl} \cdot 42}{m} \times 100\% \tag{2}$$

where $V_0$ is the volume of the HCl standard solution consumed by the blank sample, $V$ is the volume of the HCl standard solution consumed by the sample, and $m$ is the mass of the sample [52].

*2.6. Characterization of Self-Healing Coatings*

The self-healing capabilities of a dual self-healing system were evaluated through scratch tests, wear tests, and saltwater corrosion tests.

The scratch tests were performed on the surface of cured resin, where a scratch of approximately 1 cm in length was made using a scalpel, and a 1 mm mark was applied with a marker pen [53–55]. Observations were conducted under an optical microscope to ensure that the area of interest was centrally located within the field of view. Changes in the width of the scratch, before and after the self-healing process, were observed and quantitatively analyzed using Nano Measure software 1.2.

The wear tests were conducted to simulate the complex damage scenarios that are encountered in practical applications. Sandpaper was applied with a uniform force across a marked 1 cm² area, resulting in uniform damage to the resin surface. This damage was subsequently observed under an optical microscope.

The corrosion resistance of the dual self-healing system was observed through saltwater immersion experiments [56]. The experiments applied undoped epoxy resin, epoxy resin doped with microcapsules, undoped PU-F520 resin, and PU-F520 resin doped with microcapsules onto 4 cm × 2.5 cm-sized tinplate sheets. The application area was 2.5 cm × 2 cm, and each group of resin included three samples: one as a control and the other with a crosshatch scratch treatment on the surface. These samples were then immersed in 10 wt. % saltwater to observe and compare the corrosion resistance of each resin on the tinplate [57].

## 3. Result and Discussion

*3.1. Synthesis of Diamine Microcapsules*

As is shown in Figure 1, utilizing interfacial polymerization, three types of microcapsules were successfully prepared based on different diamines: EDA, HMDA, and MXDA, labeled as MCs-EDA, MCs-HMDA, and MCs-MXDA, respectively. SEM observations revealed that these microcapsules exhibited favorable morphology and uniform size distribution, with no flocculation observed (Figure 1a–c).

The average particle size for MCs-EDA was measured at 0.22 μm (Figure 1d), with a size distribution variance ($\sigma^2$) of 0.003. For MCs-HMDA (Figure 1e), the average size was 0.28 μm with a variance ($\sigma^2$) of 0.007. MCs-MXDA had an average size of 0.34 μm with a variance ($\sigma^2$) of 0.012 (Figure 1f). These variations could be attributed to the chemical properties of the different diamines, including molecular size and compatibility with heptane. Moreover, observations through optical microscopy (OM) indicated that the size of the diamine emulsion droplets was consistent with the SEM results, further confirming the uniformity of the particle size.

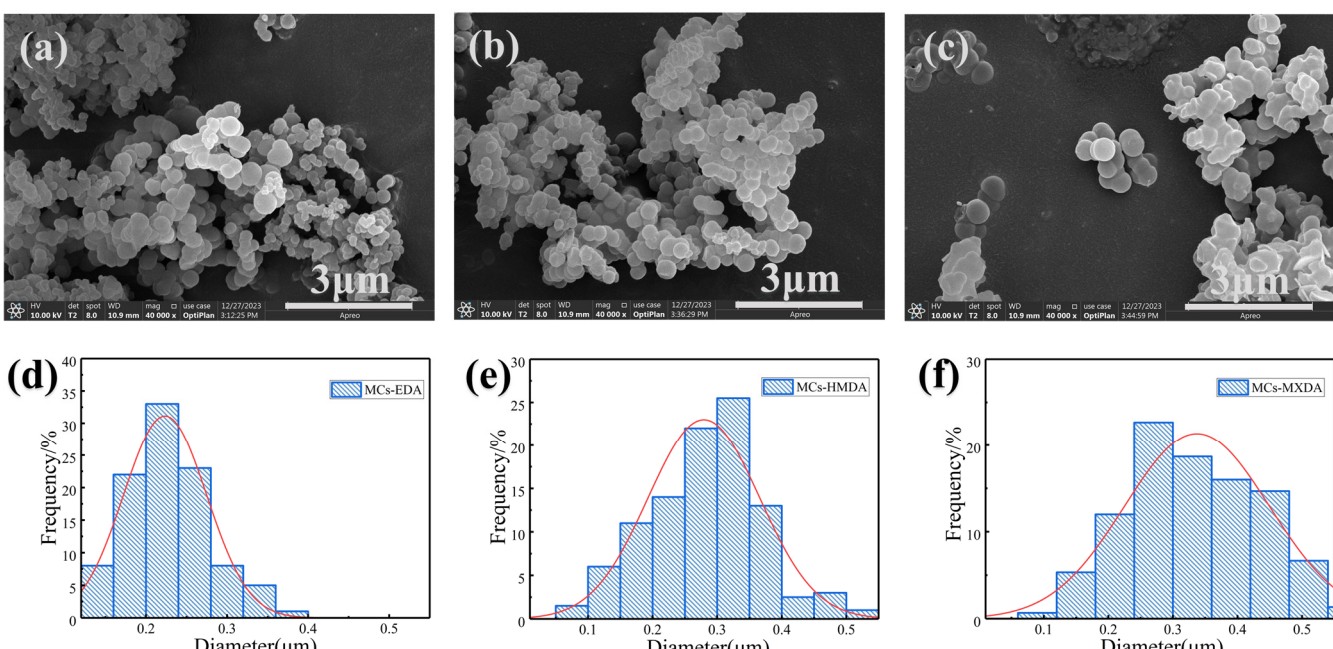

**Figure 1.** SEM image and size distribution of different diamine microcapsules, (**a**,**d**) MCs-EDA, (**b**,**e**) MCs-HMDA, and (**c**,**f**) MCs-MXDA.

It Is noteworthy that an increase in average particle size corresponded with a broader size distribution (i.e., an increase in variance $\sigma^2$). This could be due to the uneven shear forces during the emulsification process. Near the turbulent center, droplets underwent more intense emulsification, leading to smaller sizes, whereas on the outskirts of the turbulence, the droplets were larger due to lower shear forces. As the average particle size increased, this effect became more pronounced, leading to a greater dispersion in size distribution.

TEM imaging was employed to further investigate the internal structure of the microcapsules. The TEM images clearly displayed the core–shell structure of the microcapsules, with a distinct boundary and contrast between the core and the shell (Figure 2a). The particle size, measured at approximately 500 nm, was consistent with SEM images and particle size analysis results, further validating these measurements' accuracy. Additionally, the wall thickness of the microcapsules was found to be around 140 nm, which ensures the stability and functionality of the microcapsules. In the TEM images, the shell appeared uniformly dense without any apparent defects or voids, indicating that the fabrication process was refined, and the quality of the shell layer was well controlled.

Combining observations from SEM and TEM allows for the following conclusions. First, all three types of diamine-based microcapsules had a clear core–shell structure, ensuring the encapsulation of the liquid diamines and the formation of the shell layer. Second, the uniformity in particle size and wall thickness indicates a stable and controllable fabrication process, which guarantees their subsequent uniform dispersion and stable performance in resin materials, providing a solid foundation for practical applications. Lastly, the TEM observations corroborated the SEM images and particle size analysis results, enhancing our understanding of the structural characteristics of the microcapsules. These findings strongly support further application research and process optimization.

Infrared spectroscopy (IR) was employed to further verify the chemical composition and internal structure of the microcapsules synthesized from various liquid diamines. Figure 2c compares the infrared spectra of three microcapsule types: MCs-EDA, MCs-HMDA, and MCs-MXDA.

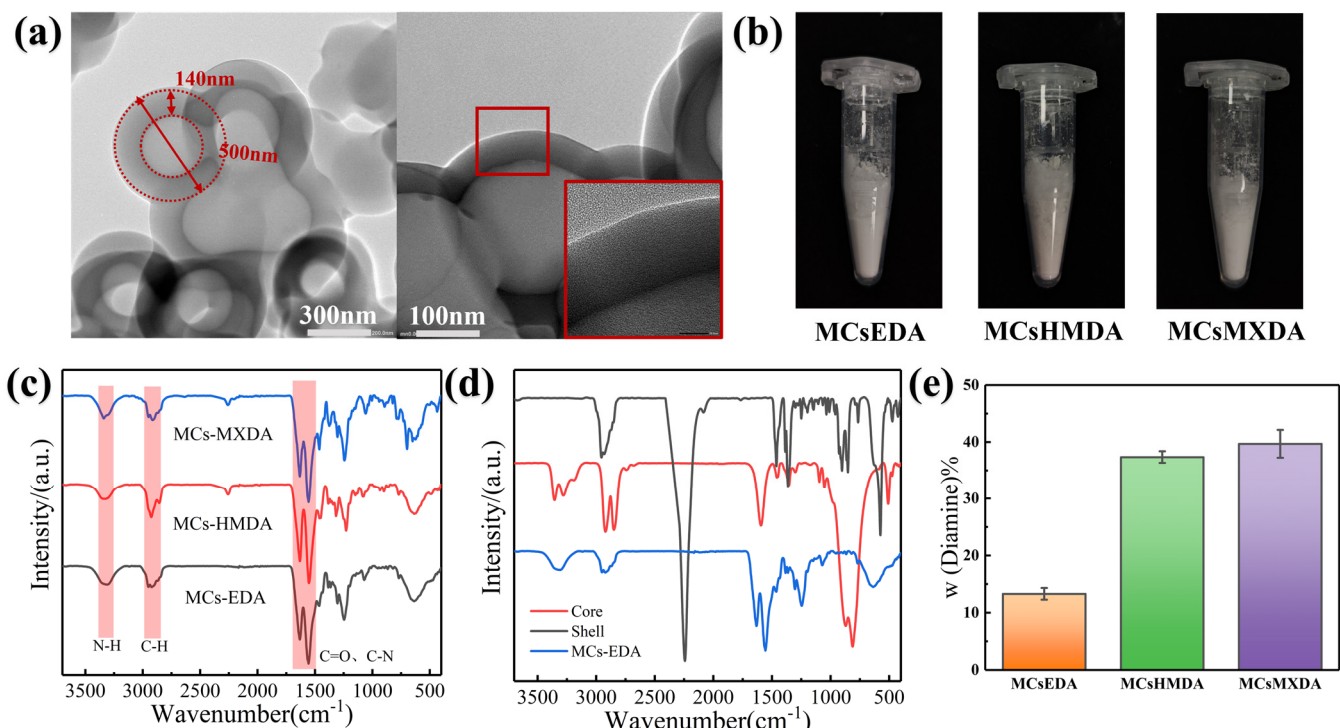

**Figure 2.** Composition analysis of diamine microcapsules. (**a**) TEM image of MCs-EDA and enlarged view of shell structure. (**b**) Photographs of different microcapsules. (**c**) FTIR spectra of different microcapsules. (**d**) FTIR spectra of core, shell materials, and MCs-EDA. (**e**) The core content of different microcapsules.

Initially, the peaks at about 3300 cm$^{-1}$, corresponding to the N–H stretching vibrations, directly prove the successful encapsulation of liquid diamines within the microcapsules. The peaks at 2900 cm$^{-1}$ correspond to the C–H stretching vibration, with the lower peak positions indicating hydrogen bonding between urea groups in the polyurea shell [54]. Furthermore, the twin peaks at 1617 cm$^{-1}$ and 1551 cm$^{-1}$, representing the absorption peaks for C=O and C–N, respectively, with the lower peak positions signifying the C=O vibration absorption of amides, confirm the formation of amide and urea bonds within the microcapsule shell. Notably, small peaks at 2259 cm$^{-1}$, characteristic of the free NCO (isocyanate) groups [55], are observed for MCs-HMDA and MCs-MXDA. The presence of these peaks suggests the possibility of unreacted isocyanate groups within these microcapsules.

Figure 2d further confirms the microcapsule shell structure and core material encapsulation by comparing the infrared spectra of the shell material, core material, and the microcapsules. The disappearance of the isocyanate group's absorption peak at 2259 cm$^{-1}$ in the microcapsules' infrared spectrum [52], as shown in the figure, indicates the successful reaction of IPDI with the liquid diamine during interfacial polymerization, resulting in a dense shell. The appearance of the amide functional group at 1617 cm$^{-1}$ further confirms the occurrence of this reaction. These results are consistent with the core–shell structure observed by TEM, thus verifying the accuracy of the SEM and TEM analyses. In summary, infrared spectroscopy analysis offers compelling evidence of the microcapsules' chemical composition and internal structure, confirming the successful encapsulation of liquid diamines within the dense shell formed by the reaction between IPDI and liquid diamine. This lays the foundation for the application of microcapsules as healing agents in resin materials, particularly for the realization of their self-healing, waterproofing, and anti-corrosion functions.

In this work, the core material content of the microcapsules was measured using titration (Figure 2e). By dissolving the shell layer with an organic solvent, the encapsulated core material is released, and the free amine groups react with hydrochloric acid. By

recording the volume of hydrochloric acid solution consumed, we can calculate the content of the diamine. The results are shown in Figure 2e. Titration results revealed that the EDA content in MCs-EDA was approximately 13.29 wt. % (2.22 mmol/g), the HMDA content in MCs-HMDA was 37.36 wt. % (3.20 mmol/g), and the MXDA content in MCs-MXDA was approximately 39.69 wt. % (2.91 mmol/g).

To further verify the successful encapsulation of liquid diamines within the microcapsules and investigate their thermal stability, thermal analysis methods such as TGA and DSC were employed for examination (Figure 3).

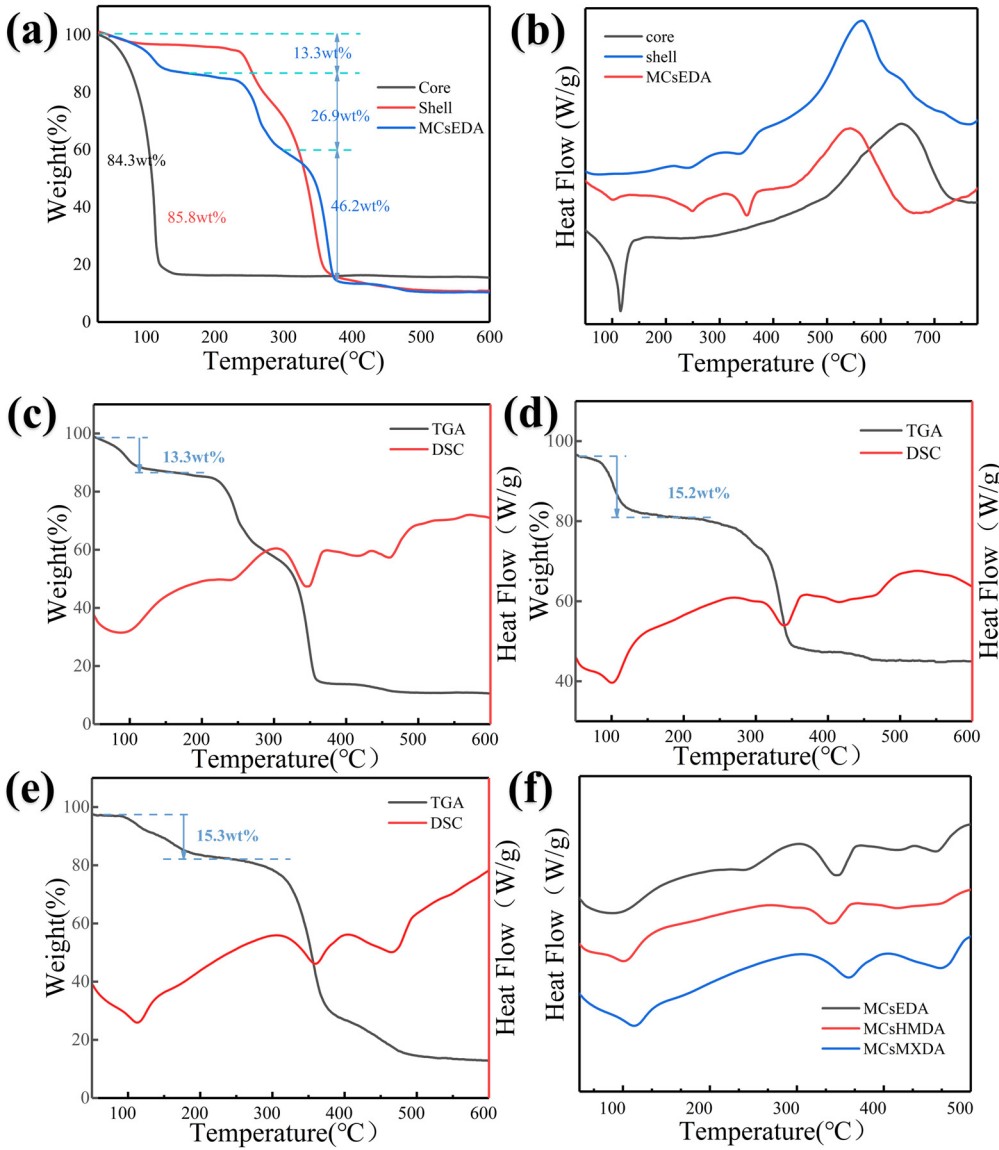

**Figure 3.** Thermal characterization of diamine microcapsules. (**a**,**b**) TGA and DSC curves of core, shell materials, and MCs-EDA. (**c**,**d**,**e**) TG-DSC curves of MCs-EDA, MCs-HMDA, and MCs-MXDA. (**f**) DSC curves of different diamine microcapsules.

Figure 3a,b shows the TGA and DSC curves of the core, shell, and microcapsule encapsulating ethylenediamine. From the TGA curves, it is likely that the first weight loss platform of MCs-EDA corresponds to the encapsulated core material, ethylenediamine. The weight loss of this platform is approximately 13.3 wt. %, which is consistent with the independently calculated core material content obtained through titration. This result is similar to the results of titration part. It not only verifies that the microcapsule successfully encapsulates the predetermined amount of ethylenediamine but also indicates that the

shell of the microcapsule can effectively maintain its integrity during thermal analysis until the core material is completely released.

It is noteworthy that the loss temperature exhibits a lag toward the high-temperature region compared to the pure core material or shell. This phenomenon suggests that the thermal stability of the core material, ethylenediamine, is significantly improved through the encapsulation by the microcapsule. This sustained-release effect not only helps to protect the core material from the direct impact of high-temperature environments but also controls the release rate of the core material at specific temperatures, thereby prolonging its service life and effectiveness. Additionally, the polymerized shell material exhibits a high decomposition temperature. This means that even under high-temperature conditions, the shell of the microcapsule can maintain its structural integrity, preventing premature leakage or the loss of the internal core material. This excellent thermal stability ensures the reliability and durability of the microcapsule in high-temperature applications.

By comparing the DSC and TGA curves (Figure 3a,b), a distinct endothermic peak is observed in the DSC curve at approximately 120 °C. This peak is attributed to the release of small molecules, particularly the liquid evaporation of the core material, ethylenediamine. Since this endothermic peak is like the endothermic peak of pure ethylenediamine (EDA), it indicates that ethylenediamine has been successfully encapsulated within the microcapsule. This finding is consistent with the results of the TGA curve, further confirming the successful preparation of the microcapsule.

Furthermore, for the microcapsule MCs-EDA, the endothermic peaks observed at 250 °C and 351 °C correspond to the DSC curve of the shell. These two endothermic peaks align with the two mass loss platforms in the TGA curve, corresponding to the two reactions during the decomposition of the polyurea resin. The results from both the TGA and DSC curves further verify the successful encapsulation of EDA and explain the excellent thermal stability and sustained-release effect of the microcapsule.

Figure 3c–e compares the TGA and DSC curves of MCs-EDA, MCs-HMDA, and MCs-MXDA synthesized using three different diamines. Observation reveals that each of these microcapsules exhibits three endothermic peaks, which correspond to the three weight loss platforms in the TGA curves. Based on previous analysis, these endothermic peaks can be attributed to the endothermic effect of liquid diamine evaporation due to small molecule evaporation and the endothermic effect of shell pyrolysis reactions. Correspondingly, in the TGA curves, the mass losses (wt. %) are as follows: 13.3% for MCs-EDA, 15.2% for MCs-HMDA, and 15.3% for MCs-MXDA. These results are generally consistent with the preliminary results of titration statistics, although the titration values were slightly higher than the mass losses determined by TGA.

This discrepancy can be attributed to chemical changes that occur during the microcapsule encapsulation process. During encapsulation, the diamine molecules react with the isocyanate material, forming not only a dense shell layer but also short-chain or diamine-terminated long-chain polymers. These polymers still exhibit some reactivity during titration analysis, leading to a higher titration result. These polymers may have higher thermal stability and decompose at higher temperatures during TGA analysis, resulting in lower mass losses at lower temperatures. Since the terminal amino groups maintain their reactivity in the self-healing reaction, we will rely on the titration results for calculations in subsequent microcapsule doping experiments to ensure the accurate metering of the diamine components within the microcapsules.

The endothermic peaks observed in Figure 3f near 105 °C exhibit a notable shift, which is particularly noteworthy. This shift is attributed to the boiling point differences among the three diamines used. Specifically, the boiling point of ethylenediamine is 116–117.3 °C, the boiling point of 1,4-cyclohexyldiamine is 199.4 °C, and the boiling point of m-xylylene diamine is 247 °C. This difference in boiling points is reflected in both the DSC and TGA curves. From the DSC curves, the first peak position increases in accordance with the increasing boiling point.

Through TGA and DSC analysis, a deeper understanding of the internal structure of microcapsules and the changes in thermal properties of microcapsules with different core materials has been gained. The results showed that the three types of microcapsules successfully encapsulated liquid diamines and exhibited excellent thermal stability, given the volatile nature and low cost of ethylenediamine, its encapsulation holds practical significance. Consequently, MCs-EDA will be further explored in subsequent studies.

### 3.2. Synthesis of IPDI Microcapsules

Similar to the system employed for synthesizing diamine microcapsules, the preparation of isocyanate microcapsules involved the use of water as the dispersing phase and Tween80 as the emulsifier (Figure 1). By the dropwise addition of the oil-in-water emulsion into a solution containing dissolved diamine, followed by curing, washing, filtering, and drying, isocyanate microcapsules were successfully obtained.

To investigate the influence of the catalyst (curing agent) during interfacial polymerization, three different diamines were selected: ethylenediamine, diethylenetriamine, and polyetheramine D230. SEM imaging revealed that a regular spherical structure with good dispersibility and minimal aggregation could be achieved through interfacial polymerization (Figure 4). Among the microcapsules synthesized using different diamines, MCs-IPDI(D230) exhibited the best morphology, followed by MCs-IPDI(DETA). However, MCs-IPDI(EDA) showed wrinkles on its surface and some degree of aggregation (Figure 4a). This variation can be attributed to the different chemical properties of the three diamine curing agents. Ethylenediamine, with its high alkalinity, might have influenced the surface morphology of the microcapsules. Additionally, its shortest chain length might have resulted in insufficient polymerization on the microcapsule surface, leading to its fragility. Diethylenetriamine, as a triamine, enhances the crosslinking density of polyurea on the microcapsule surface, forming a dense shell (Figure 4b). Its rapid reaction rate, however, could lead to the formation of some polyurea impurities, evident in the SEM images. Polyetheramine D230, with its longer chain length, produced a denser shell compared to the other two diamines but required a longer reaction time for curing due to its lower reactivity (Figure 4c). The particle size analysis (Figure 4d–f) of the three MCs-IPDI revealed average particle diameters of 1.68 μm for MCs-IPDI(EDA) with a variance ($\sigma^2$) of 0.56, 0.93 μm for MCs-IPDI(DETA) with a variance ($\sigma^2$) of 0.23, and 1.71 μm for MCs-IPDI(D230) with a variance ($\sigma^2$) of 0.67. The rapid reaction rate of DETA resulted in smaller particle sizes, especially when the droplets were small, leading to a more significant distribution of smaller particles compared to the other two diamines.

TEM further characterized the core–shell structure and encapsulation of the IPDI microcapsules (Figure 5a–c). All three microcapsules exhibited a distinct core–shell structure. While MCs-IPDI(EDA) showed some degree of aggregation, the core–shell boundary was still evident. The core–shell structure of MCs-IPDI(DETA) and MCs-IPDI(D230) was more pronounced, confirming the successful encapsulation of IPDI.

To determine the core content of the isocyanate microcapsules, the method used to measure the isocyanate group content in polyurethane prepolymers was referenced. The isocyanate group content (wt. %) was used as a metric (Figure 5d). The results showed NCO% values of 1.02% (0.24 mmol/g), 10.05% (2.39 mmol/g), and 5.41% (1.29 mmol/g) for the three microcapsules, respectively. The higher reaction rate of DETA resulted in a higher NCO content as it rapidly reacted with and encapsulated the IPDI.

Through IR analysis, a comprehensive understanding of the characteristics of the three MCs-IPDI can be achieved (Figure 5f). Particularly noteworthy is the region around 2256 cm$^{-1}$ [52,53], which corresponds to the characteristic peak of isocyanate (NCO). At this position, MCs-IPDI(EDA) exhibits almost no discernible peak, suggesting a low content of isocyanate in this compound or some form of interaction that affects the NCO groups. In contrast, both MCs-IPDI(DETA) and MCs-IPDI(D230) exhibit pronounced peaks at 2256 cm$^{-1}$, confirming the presence of isocyanate in these two compounds. Notably, the peak intensity of MCs-IPDI(D230) is lower than that of MCs-IPDI(DETA). These ob-

servations align with previous titration results, further emphasizing the consistency and reliability of the experimental data.

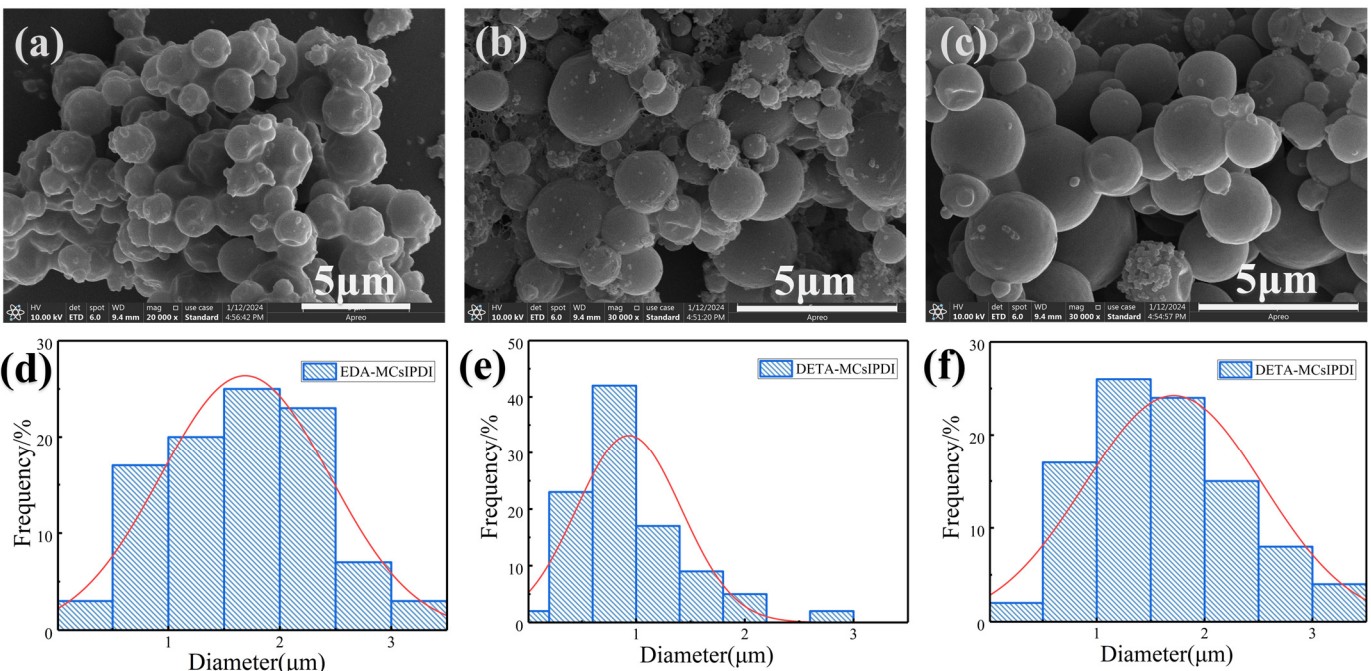

**Figure 4.** SEM image and size distribution of different IPDI microcapsules, (**a**,**d**) MCs-IPDI(EDA), (**b**,**e**) MCs-IPDI(DETA), and (**c**,**f**) MCs-IPDI(D230).

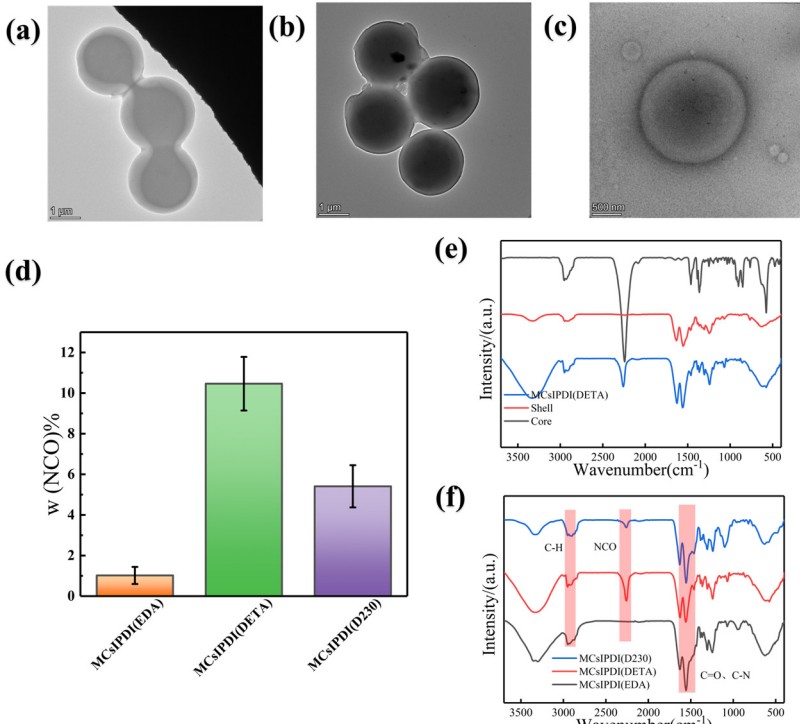

**Figure 5.** Composition analysis of IPDI microcapsules. TEM image of (**a**) MCs-IPDI(EDA), (**b**) MCs-IPDI(DETA), and (**c**) MCs-IPDI(D230); (**d**) core content of MCs-IPDI; (**e**) FTIR spectra of core, shell materials, and MCs-IPDI(DETA); (**f**) FTIR spectra of different MCs-IPDI.

To further validate the successful encapsulation of IPDI, we compared the IR spectra of the core material, shell material, and MCs-IPDI(DETA) (Figure 5e). Through this com-

parison, it is evident that the characteristic peaks of IPDI are present in MCs-IPDI(DETA), providing conclusive evidence for the successful encapsulation of IPDI through interfacial polymerization. Additionally, the broad peak in the range of 3000–3500 cm$^{-1}$ can be attributed to the formation of amide functional groups and hydrogen bonding.

The thermal stability of the microcapsules was further observed based on TGA and DSC data (Figure 6). As shown in the figures, the thermogravimetric curves of the microencapsulated MCs-IPDI exhibit only one weight loss platform, and correspondingly, the DSC curves also display a single endothermic peak. This phenomenon is observed in all three MCs-IPDI compounds: MCs-IPDI(EDA), MCs-IPDI(DETA), and MCs-IPDI(D230) (Figure 6a–c). Additionally, an increase in the encapsulation amount of isocyanate leads to an increase in the slope of the TGA weight loss curve (Figure 6e). This may be attributed to the overlap of the thermal decomposition temperature of IPDI with the shell layer after microencapsulation. As more IPDI occupies a larger space inside the microcapsules, more IPDI and shell material decompose simultaneously when the temperature rises, resulting in an accelerated weight loss rate and an increased slope of the weight loss curve. This overlap of two endothermic peaks at similar temperatures is observed in the DSC curves, supporting this theory (Figure 6f).

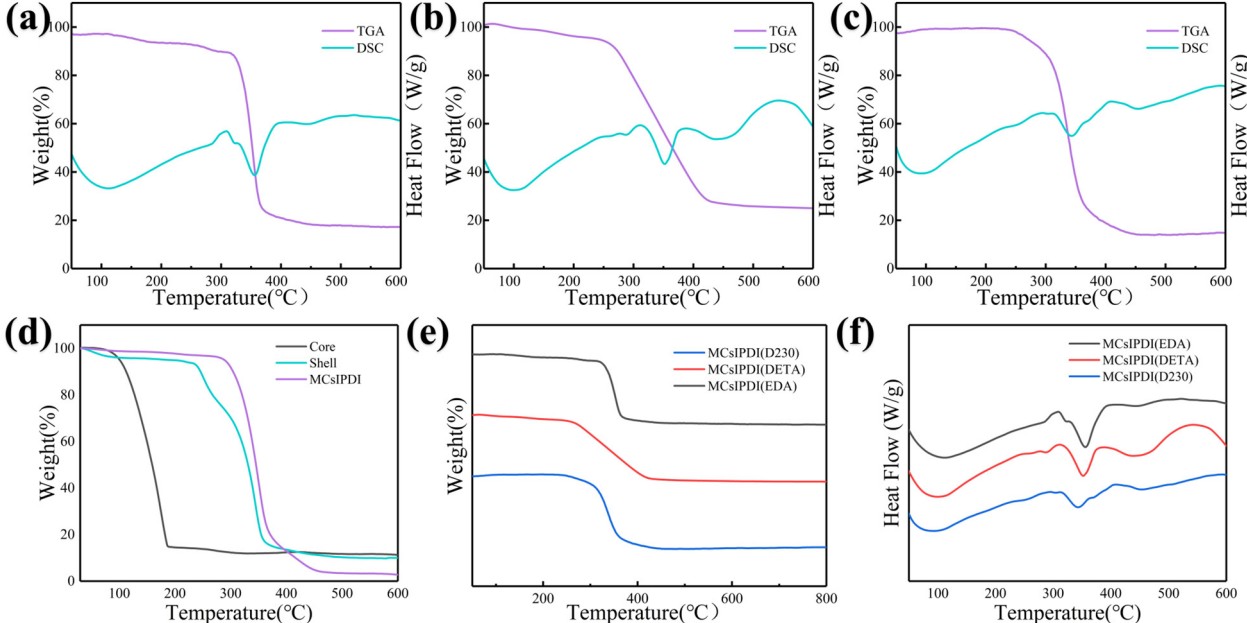

**Figure 6.** Thermal characterization of IPDI microcapsules. (**a**–**c**) TGA and DSC curves of MCs-IPDI(EDA), MCs-IPDI(DETA), and MCs-IPDI(D230). (**d**) TGA curves of core, shell materials, and MCs-IPDI(DETA). (**e**) TGA curves of different IPDI microcapsules. (**f**) DSC curves of different IPDI microcapsules.

### 3.3. Self-Healing Properties of Dual Microcapsules System

To comprehensively evaluate the self-healing properties of the resin doped with the dual microcapsule system, scratch and abrasion tests were employed. These experiments encompassed simulations of different scratch types and depths, as well as abrasion tests and saltwater experiments to mimic potential damage scenarios encountered in practical usage.

In the scratch experiment, we maintained consistent scratch depth and speed to control variables [54,55] and ensure reliable experimental results (Figure 7). By observing and analyzing the scratched areas, we could visually assess the self-healing effectiveness of the resin after scratching. It was evident that the epoxy resin doped with microcapsules, after 48 h of self-healing, reduced a scratch width of approximately 0.32 mm to 0.25 mm,

resulting in a self-healing efficiency of approximately 21.8% (Figure 7a). The calculation method is as follows:

$$\eta = \frac{l_0 - l}{l_0} \times 100\% \tag{3}$$

where $\eta$ represents the self-healing efficiency, $l_0$ is the initial width of the scratch, and $l$ is the final width of the scratch.

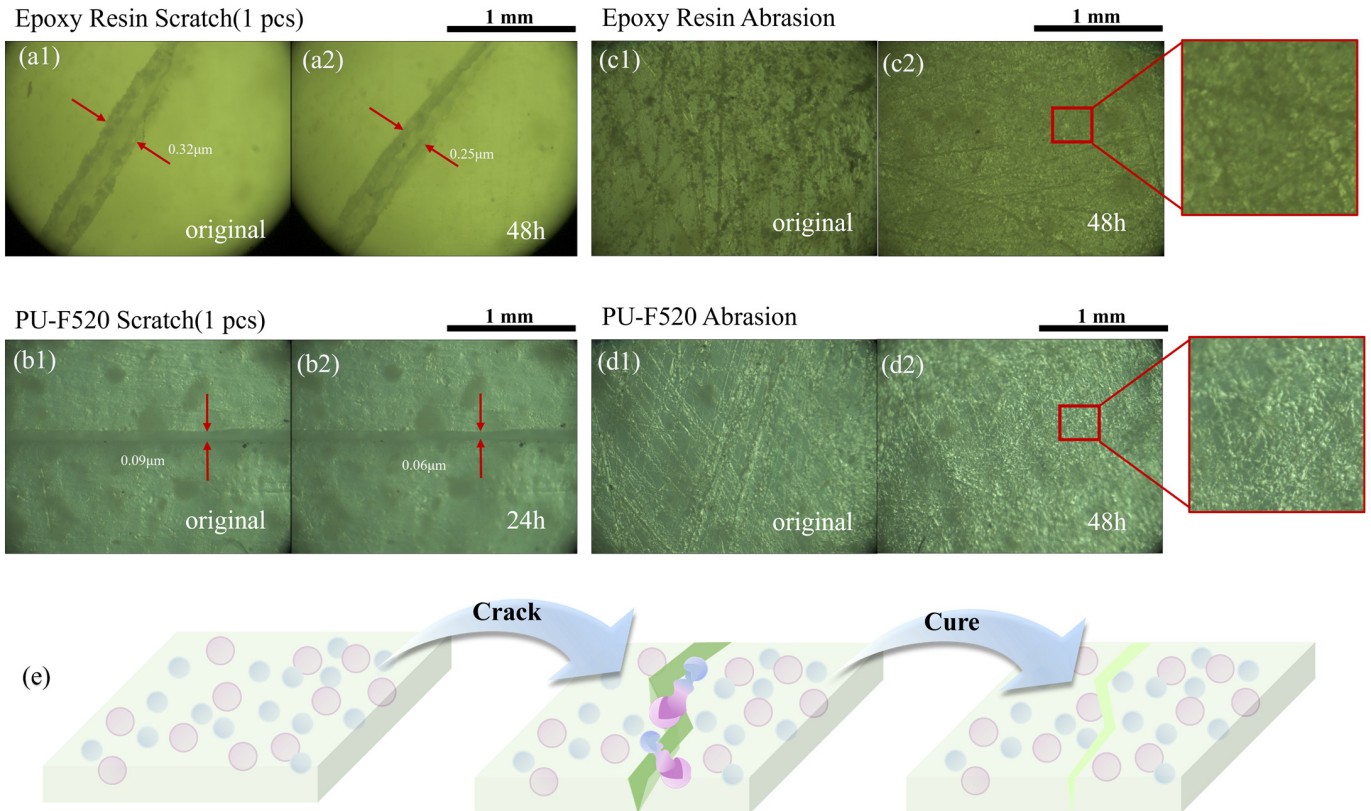

**Figure 7.** The OM image of scratched coating. (**a**) Epoxy resin scratch. (**b**) PU-F520 scratch. (**c**) Epoxy resin abrasion. (**d**) PU-F520 abrasion. (**e**) The mechanism of crack repairing.

Similarly, for the PU-F520 resin, it was observed that the scratch width narrowed from an initial 0.09 μm to 0.06 μm after 24 h of self-healing, with the change being visibly noticeable to the naked eye, and the self-healing efficiency was determined to be 33.3%. Ma et al. [56] conducted scratch experiments, and by using SEM, the release and curing of healing agents near the scratches were observed. The significant release and repair of healing agents were also observed by Hu et al. [50]. Compared with these studies, the microcapsules prepared in this work, with smaller particle size, are conducive to the targeted repair of fine scratches and exhibit a more pronounced self-healing efficiency on a macroscopic level.

The abrasion test is an experimental method that simulates wear and tear on the surface of the resin (Figure 7c,d). By simulating abrasions using sandpaper on the resin surface, we can observe the smoothness of the resin after frictional damage and assess its self-healing properties. This experimental approach is crucial for evaluating the self-healing performance of resins in natural environments. As shown in Figure 7c,d, the originally continuous and distinct striped scratches became discontinuous after self-healing, with significant reductions in scratch width and depth. Some scratches even disappeared almost completely. These changes indicate that after abrasion damage, the self-healing agents within the microcapsules have successfully released and effectively repaired the resin surface. As a key component of the dual microcapsule system, the self-healing agents within respond rapidly when the resin surface sustains abrasion damage, filling the voids

created by the scratches (Figure 7e). As the self-healing agents solidify and stabilize, the morphology of the scratches improves significantly, effectively restoring the continuity and smoothness of the resin surface. This self-healing process not only enhances the wear resistance and durability of the resin but also provides assurance for its long-term application in complex environments.

### 3.4. Anticorrosion Ability

In addition, a saltwater immersion test was conducted to comprehensively evaluate the self-healing properties of the resin doped with the dual microcapsule system and its durability under different environmental conditions [57,58]. Aside from the previously mentioned single scratch and abrasion tests, a saltwater immersion test was conducted to simulate the corrosion and aging issues that the resin may encounter in practical applications, such as when immersed in seawater. The salt and oxygen in the saltwater can react with the resin, leading to a decrease in its performance. By observing and analyzing the surface characteristics of the resin after saltwater immersion, we can understand its corrosion resistance and anti-aging properties.

The corrosion resistance of the self-healing system was observed through saltwater immersion experiments. As is seen in Figure 8a, the epoxy resin coating without doping showed significant corrosion and yellowing on the surface after 10 days, while the MCs-doped epoxy resin coating maintained a consistent color with no apparent rust corrosion. Similarly, PU-F520 resin, after the same treatment, exhibited noticeable rust at the edges of the undoped coating, with slight surface yellowing, whereas the MCs-doped resin coating demonstrated enhanced corrosion resistance, with almost no change in surface color, and even exhibited self-healing effects on scratches.

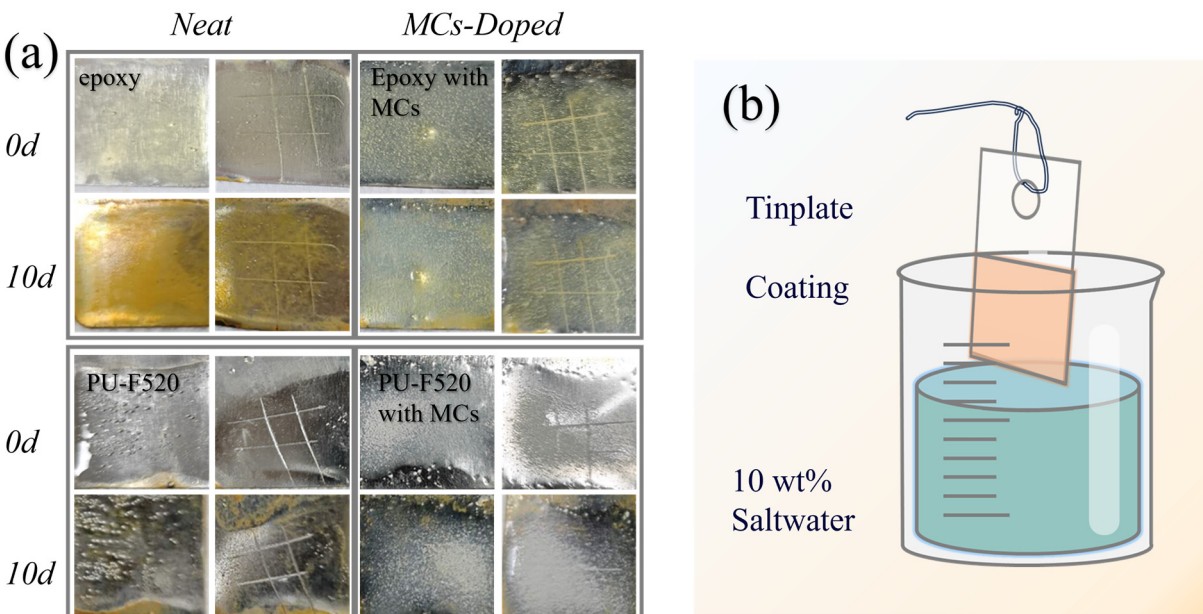

**Figure 8.** Anticorrosion ability of dual MCs system. (**a**) Photographs of coatings. (**b**) Experimental schematic of saltwater corrosion resistance of tinplate coatings.

The results indicate that the microcapsule-doped resin systems exhibit more pronounced corrosion resistance. This could be attributed to the introduction of an additional protective layer by the microcapsules, effectively isolating the resin from corrosive agents in the external environment such as moisture, oxygen, and salt, thereby reducing the risk of corrosion [59,60]. Second, microcapsules can promptly release self-healing agents to repair minor damage on the resin surface. This self-healing mechanism helps restore the integrity and continuity of the resin surface, filling potential corrosion channels or defects during

the healing process, further enhancing the corrosion resistance of the resin. Figure 8b is the experimental schematic of the saltwater corrosion resistance of tinplate coatings.

Furthermore, the incorporation of microcapsules can also improve the physical and chemical properties of the resin system, such as increasing its compactness, reducing porosity, and enhancing chemical corrosion resistance. These improved properties contribute to enhancing the overall durability of the resin, making it more resilient to harsh environmental conditions.

## 4. Conclusions

In this research, we developed a dual microcapsule system composed of diamine and IPDI through interfacial polymerization, employing n-heptane as an emulsifying agent for anhydrous microencapsulation. We synthesized three varieties of polyurea-shelled liquid diamine microcapsules under controlled conditions, achieving up to 39.69 wt. % diamine content, characterized by their highly ordered spherical morphology and stability. The encapsulation efficacy and microcapsules' performance were confirmed via infrared spectroscopy, titration, and thermal analysis.

Similarly, IPDI microcapsules were produced, exploring the impact of various diamines and curing agents on encapsulation efficiency. These microcapsules, particularly those synthesized with high cross-link density diamines, demonstrated uniformity in size, spherical integrity, and a notable NCO group content of 10.05%, indicating their chemical robustness and applicability.

The integration of this microcapsule system into different resin matrices led to significant enhancements in their physical and chemical attributes, including density improvement, porosity reduction, and chemical resistance augmentation. Notably, the system endowed the resins with effective self-healing capabilities, facilitating damage recovery and service life extension, as evidenced by self-healing efficiencies of 21.8% and 33.3% in epoxy and PU-F520 resins, respectively. The system's versatility was further demonstrated in various damage scenarios, such as scratches, abrasion, and seawater corrosion, where the microcapsules actively released healing agents to repair the resin surfaces.

This study presents a dual microcapsule system with potential implications for improving material durability and functionality across multiple industries, including automotive, aerospace, and construction. It contributes to the field by offering a practical approach to enhancing material resilience and longevity through self-healing and corrosion resistance features.

**Author Contributions:** Conceptualization, methodology, writing—original draft preparation, H.M.; formal analysis, investigation, Y.D., W.Z. and X.Y.; writing—review and editing, Q.Z. All authors have read and agreed to the published version of the manuscript.

**Funding:** The authors declare that this study received funding from Sichuan University–Carpoly Chemical Group Co., Ltd. Cooperation Project (No. 22H1403). The funder was not involved in the study design, collection, analysis, interpretation of data, the writing of this article or the decision to submit it for publication.

**Institutional Review Board Statement:** Not applicable.

**Informed Consent Statement:** Not applicable.

**Data Availability Statement:** The data presented in this study are available on request from the corresponding author.

**Conflicts of Interest:** The authors declare that they have no known competing financial interests or personal relationships that could have appeared to influence the work reported in this paper.

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
