# Peer review of "Dual Microcapsules Encapsulating Liquid Diamine and Isocyanate for Application in Self-Healing Coatings"

_coatings, doi:10.3390/coatings14040410_

Round 1

Reviewer 1 Report

Comments and Suggestions for Authors

The authors developed dual microcapsules encapsulating liquid diamine and isocyanate for self-healing coatings.The coating material showed high self-repair efficiency. The discussion part of the manuscript is quite weak. There are no references in the results and discussion part. The manuscript should be reconsidered considering this situation. I believe that the following problems must be corrected before the manuscript can be published.

- It would be better if the relevant peaks in the FTIR spectrum are also shown on the figure. The references of the peaks indicated in the FTIR should also be added.

- The details about the scratch and abrasion tests should be given in the experimental part. Some of what is written in the result and discussion part for both tests should be moved to the experimental part.

- Did the authors measure the viscosity of the self-healing agents?

- The self-healing efficiency values of the study should be compared with the values obtained from similar studies in the literature.

- Scheme 1. should be moved to experimental part.

- Reference 15, given in line 37, has no parenthesis.

Author Response

Thank you for your detailed and constructive criticism, the corresponding responses can be found in the PDF document.

Reviewer 2 Report

Comments and Suggestions for Authors

Referee´s comments

To the paper entitled ``Dual Microcapsules Encapsulating Liquid Diamine & Isocyanate for Application in Self-Healing Coatings`` by Huaixuan Mu et al.

The paper is devoted to the development microcapculating method.  Capsules, a size of which is less than 1 micron or about 2 microns, have been obtained and investigated. The capsules are suggested to be self-healing.

Of course, the paper should be published earlier or later, but a revision is recommended.

Introduction. Please mention an application field of microcapsules. For which purpose did you develop them? As  follows from the conclusions, they are a component of anticorrosion coating. Explain a principle: microcapsules are added to coating, their content is released (how?), coating is self-reconstructed.  

Experimental. Please avoid subsubsections like 2.5.1 etc. Otherwise the paper looks like a report.  It is the same for the section Results and discussion.

Formulas must be numbered. It is the same regarding the formula for self-healing efficiency.

Results and discussion

Polymerization reactions are very welcome. Show chemical nature of the self-reconstruction of coating. It is also unclear how chemical substances are released from microcapsules.

Recommendations for a future: electrochemical investigation of corrosion is very welcome.

Comments on the Quality of English Language

English must be slightly improved

Author Response

(The authors gave the same response as above.)

Reviewer 3 Report

Comments and Suggestions for Authors

I carefully reviewed this manuscript. There are too many corrections and modifications for acceptance as follows.

More explanations and descriptions are added on experimental results and findings obtained in this study with some numerical data in Abstract.

L9: The use of possessive form such as “polyurea’s” should be avoided.

L28: The representation of “not only due to A but also due to B” or “not only due to A but also to B” is better.

L37: “extrinsic15.” is changed to “extrinsic [15].”

L97-193: One or more references should be cited in each section of 2.1-2.5.3. I'm feeling something wrong with no references in the experimental section.

L116: The term of “hours” is changed to “h” or “hr”. The same goes for others.

L125: “Diamine is dissolved” Here, which diamine was used?

L176: A space is inserted between “25” and “ml”. The same goes for others.

L184: The symbol “NCO%” is not defined, or not explained.

L186: The term of “CHCl” is already defined. This sentence is deleted.

L197: The experimental conditions for preparation of microcapsules are required.

L199: The amine compounds are already abbreviated.

L200: SEM is already abbreviated.

L197: The results obtained from Figure 1 are not described and not discussed.

L204: Information of the particle size is not obtained from Figure 2e.

L206: Figure 2f is shown in this manuscript.

Figure 1: The scale of x axis should be unified. It is unclear as it is.

L221: TEM is already abbreviated.

L224: I can’t read the particle size of 350 nm and thickness of 79 nm form Figure 2a because the picture is unclear and the scale is not shown.

L232: At least, I can’t see a core-shell structure.

L241-239: Although IR measurements of diamine microcapsules are performed, I think more instrumental results or evidence on the presence of amino groups are required.

L275-282: It is difficult to understand of the content of amino groups from w(diamine)%. The content is shown in for example mmol/g.

L291: TGA and DSC are already abbreviated.

Figure 3: The scale of x axis should be unified.

L326: TGA is already abbreviated.

L327: DSC is already abbreviated.

L354: The abbreviation should be used.

Figure 4: The scale of x axis should be unified.

L401-406: It is difficult to understand of the content of NCO groups from w(NCO)%. The content is shown in for example mmol/g. In addition, why is the NCO content determined by titration with HCl? NCO group is not ionic.

L411: IR is already abbreviated.

L431: DSC is already abbreviated.

L459: What is “l_0”?

L484-501: The descriptions on saltwater immersion test should be transferred to the experimental section.

L502: The author(s) is required to explain a expected mechanism of corrosion and color change.

Through the whole manuscript, explanations and descriptions on the findings obtained from this investigation are weak or small. More descriptions are required.

There are no references in Results and Discussion.

The description on reference #1-54 doesn’t follow by the instructions of this journal. The journal name is abbreviated in italics. Year is in bold. Volume and page range (first and last pages) are not shown. DOI is shown in “DOI:10.”.

Comments on the Quality of English Language

The quality of English is moderate. However, author(s) should carefully review this manuscript before submission. Details are shown in reviewer comment. 

Author Response

(The authors gave the same response as above.)

Round 2

Reviewer 1 Report

Comments and Suggestions for Authors

I am pleased with the authors' responses and the improvements they have made to the manuscript. I believe that the revised manuscript can be published.

Author Response

Thank you for your positive feedback on the revisions and improvements made to our manuscript. Your acknowledgment of our efforts to address the concerns raised is highly appreciated. We sincerely thank you for your time, expertise, and constructive guidance throughout the review process.

Reviewer 3 Report

Comments and Suggestions for Authors

I reviewed this revised manuscript, in particular, the sentences in red. There are some revisions. However, if these are revised, I accept to publish this manuscript in MDPI Journal, coatings.

L64: “IPDI and diamine rapidly react to” is changed to “IPDI rapidly react with diamine to”.

The latter expression is common.

L88-89: “a novel microencapsulation ‥‥ isophorone diisocyanate (IPDI)” This subject is too long. isophorone diisocyanate is deleted because it is already abbreviated.

L119: I can understand “room temperature”. On the other hand, I have not seen “room pressure”. Therefore, “and pressure” should be deleted. It is the same as L135.

L131: The order of magnitude is too small. Therefore, “9 mmol” or “9.0 mmol” is favorable.

L195: A white space is inserted between per and [55-57].

L381: “SEM” is change to for example “SEM image” or “SEM observation”.

L409 and 410: “Infrared spectroscopy” represents a method. The term of “FT-IR spectra” should be used.

P13-14: I can’t find reference number [58] in the text.

Comments on the Quality of English Language

I can smoothly read this manuscript.

Author Response

Thank you for your careful review of the revised manuscript and improvements made to our manuscript. We are grateful for the opportunity to refine our work further and for your constructive comments, which has been invaluable in improving the quality of our manuscript. Your acknowledgment of our efforts to address the concerns raised is highlyappreciated. We sincerely thank you for yourtime,expertise,and constructive guidancethroughout the review process.
